# A Time Series is Worth Five Experts: Heterogeneous Mixture of Experts for Traffic Flow Prediction

## Abstract

Accurate traffic prediction faces significant challenges, necessitating a deep understanding of both temporal and spatial cues and their complex interactions across multiple variables. Recent advancements in traffic prediction systems are primarily due to the development of complex sequence-centric models. However, existing approaches often embed multiple variables and spatial relationships at each time step, which may hinder effective variable-centric learning, ultimately leading to performance degradation in traditional traffic prediction tasks. To overcome these limitations, we introduce variable-centric and prior knowledge-centric modeling techniques. Specifically, we propose a Heterogeneous Mixture of Experts (TITAN) model for traffic flow prediction. TITAN initially consists of three experts focused on sequence-centric modeling. Then, designed a low-rank adaptive method, TITAN simultaneously enables variable-centric modeling. Furthermore, we supervise the gating process using a prior knowledge-centric modeling strategy to ensure accurate routing. Experiments on two public traffic network datasets, METR-LA and PEMS-BAY, demonstrate that TITAN effectively captures variable-centric dependencies while ensuring accurate routing. Consequently, it achieves improvements in all evaluation metrics, ranging from approximately 4.37% to 11.53%, compared to previous state-of-the-art (SOTA) models. The code will be released upon acceptance.

## 1 Introduction

Traffic prediction involves forecasting future traffic conditions based on historical data collected from sensors(Jiang et al., 2021), a task that has garnered significant attention in recent years(Jin et al., 2024). Highly accurate traffic predictions can provide valuable guidance to decision-makers, enhance safety and convenience for citizens, and reduce environmental impact(Jin et al., 2022; Cai et al., 2020). Moreover, with the rapid advancement of artificial intelligence, autonomous driving technologies will also benefit from precise and timely traffic flow forecasts(Guo & Jia, 2022).

Traffic data is predominantly spatio-temporal(Yuan & Li, 2021), indicating that it is intrinsically linked to the spatial location of sensors while also exhibiting temporal variations, thereby demonstrating considerable spatio-temporal heterogeneity. This characteristic makes certain methods that excel in conventional time series forecasting, such as Support Vector Regression (SVR)(Awad et al., 2015), Random Forest (RF)(Rigatti, 2017), and Gradient Boosting Decision Trees (GBDT)(Ke et al., 2017), less effective in spatio-temporal prediction tasks. In recent years, the latest advancements in spatio-temporal forecasting have witnessed the rise of Graph Neural Networks (GNNs) as powerful tools for modeling non-Euclidean spaces(Wu et al., 2020a; Zhou et al., 2020). Following this direction, these methods can generally be categorized based on how the graph is defined: models based on predefined spatio-temporal graphs(Yu et al., 2018), models uti-

lizing learnable graph embeddings(Wu et al., 2019; Xu et al., 2023), and models with time-varying graph structures(Hong et al.; Lee et al., 2022).

Early spatio-temporal models utilized predefined graphs based on Tobler's First Law of Geography (Miller, 2004) or on similarities in temporal proximity and spatial attributes (e.g., Points of Interest, POI) (Horozov et al., 2006; Liu et al., 2020) to model the relationships between nodes. While these graphs provide valuable prior knowledge, they often fail to capture the true dependencies between nodes, and incomplete data links can lead to omitting critical relationships. GraphWaveNet (Wu et al., 2019) addressed these limitations by introducing learnable graph embeddings, inspiring models such as RGDAN (Fan et al., 2024) and MTGNN (Wu et al., 2020b). MegaCRN (Hong et al.) further improved performance by leveraging memory networks for graph learning, building on the concept of learnable graph embeddings. However, these models remain largely sequence-centric, which limits their ability to effectively learn variable-centric representations, ultimately affecting their performance in traffic prediction. Recent studies have recognized this limitation and highlighted the importance of variable-centric representations. For instance, Gao et al. (2024) attempted to balance sequence- and variable-centric modeling through pretraining. While this approach has proven effective, simply combining sequence-centric and variable-centric modeling with weighted averages struggles to adapt to the complexities of real-world scenarios.

The concept of Mixture of Experts (MoE) was first proposed by Jacobs et al. (1991) and has since undergone extensive exploration and advancementsShazeer et al. (2017b); Yuksel et al. (2012). In the classical MoE model, multiple experts with identical architectures are employed, and sparse gating is utilized for coarse routing. Applying MoE to spatio-temporal prediction tasks allows the simultaneous use of numerous spatial modeling techniques. Lee & Ko (2024) undertake a preliminary endeavor to apply the MoE model to spatio-temporal sequence prediction, combining the strengths of various experts to model both repetitive and non-repetitive spatio-temporal patterns jointly. The MoE routing process can be viewed as a memory query, which introduces a challenge: the memory is not properly initialized at the early stages of training. This hinders MoE from learning meaningful relationships between inputs and outputs, particularly for models with significantly different structures, making it harder to train effectively.

This paper proposes a MoE model named TITAN, specifically designed for traffic flow prediction. TITAN comprises three different types of experts and a routing mechanism: 1) three sequence-centric prediction experts, 2) a variable-centric prediction expert, and 3) a prior knowledge-centric leader expert. Sequence-centric experts focus on learning temporal dependencies and capturing patterns over time, while variable-centric experts emphasize cross-variable relationships, ensuring a more comprehensive understanding of the data. We employ a low-rank matrix to align the knowledge across experts to address the challenge of integrating these heterogeneous experts. Additionally, to mitigate suboptimal routing decisions early in training, we introduce a leader expert who supervises the routing process, ensuring more informed decisions during uncertain situations. Through this adaptive routing mechanism, TITAN is capable of effectively modeling spatio-temporal data. Our main contributions are summarized as follows:

- We propose TITAN, a novel heterogeneous mixture of experts model that utilizes both sequence-centric and variable-centric experts for spatio-temporal prediction, with a leader expert supervising the routing process to improve the modeling of complex dependencies.

- We integrate models with different backbone networks into the MoE framework using low-rank adaptive matrices, effectively reducing the inductive bias inherent in traditional MoE models. This approach provides a flexible foundation for designing more complex MoE architectures.

- We design an expert annealing strategy for MoE's memory query process, gradually reducing the supervision from the leader expert, allowing TITAN to avoid suboptimal routing decisions early in training and enhance adaptability.

- Our model is evaluated on two real-world datasets, achieving improvements of 4.37% to 11.53% (average 9%) compared to state-of-the-art models.

## 2 RELATED WORK

### 2.1 TRAFFIC FLOW PREDICTION

Traffic Flow Prediction tasks exhibit significant spatio-temporal heterogeneity and complex variable interaction patterns. Traditional machine learning approaches like Support Vector Regression (SVR)(Awad et al., 2015), Random Forest (RF)(Rigatti, 2017), and Gradient Boosting Decision Trees (GBDT)(Ke et al., 2017), which rely heavily on feature engineering, struggle to capture these intricate interactions. Early spatio-temporal prediction models primarily focused on incorporating spatial information into models through graph structures that could effectively handle non-Euclidean spaces. For example, in 2018, DCRNN(Li et al., 2018) was introduced, injecting graph convolutions into recurrent units, while Yu et al. (2018)combined graph convolutions with Convolutional Neural Networks (CNN) to model spatial and temporal features, achieving better performance than traditional methods like ARIMA(Shumway et al., 2017). Although these approaches were effective, they depended significantly on predefined graphs based on Euclidean distance and heuristic rules (such as Tobler's First Law of Geography(Miller, 2004)), overlooking the dynamic nature of traffic (e.g., peak times and accidents). Later work, such as GraphWaveNet(Wu et al., 2019), addressed this limitation by constructing learnable adjacency matrices using node embeddings for spatial modeling. Despite achieving improved results, its ability to capture anomalies remained limited. More recent models like MegaCRN(Jiang et al., 2023b) improved the model's adaptability to anomalies by integrating meta-graph learners supported by meta-node libraries into GCRN encoder-decoder structures. While these models enhanced robustness, they were constrained by independent modeling techniques. This limitation has led to growing interest in spatio-temporal prediction models based on MoE structures(Jawahar et al., 2022; Zhou et al.). For instance, TESTAM(Lee & Ko, 2024) integrated three experts with different spatial capturing modules to improve spatio-temporal forecasting performance. However, these studies remain sequence-centric, limiting their ability to capture inter-variable relationships effectively. In this paper, we aim to address this challenge by jointly modeling both sequence-centric and variable-centric dependencies, allowing for adaptive consideration of both temporal and cross-variable interactions. This approach provides a more comprehensive view of the data and enhances the ability to model complex spatio-temporal dynamics.

### 2.2 MIXTURE OF EXPERTS

The Mixture of Experts (MoE) model, originally proposed by Jacobs et al. (1991), allows individual experts to independently learn from subsets of the dataset before being integrated into a unified system. Building on this concept, Shazeer et al. (2017a) introduced the sparse gated Mixture of Experts (SMoE), which utilizes a gating network for expert selection and implements a top-K routing strategy, selecting a fixed number of experts for each input. Lepikhin et al. (2020) advanced this with Gshard, and Wang et al. (2024) further demonstrated that not all experts contribute equally within MoE models, discarding less important experts to maintain optimal performance. Despite these advancements, MoE models face challenges in spatio-temporal tasks. The early training phase often leads to suboptimal routing, especially when encountering unpredictable events. In such cases, MoE struggles to query and retrieve the appropriate information from memory, resulting in ineffective routing decisions. While SMoE introduces inductive bias through fine-grained, location-dependent routing, it primarily focuses on avoiding incorrect routing and neglects to optimize for the best paths. Similarly, TESEAM(Lee & Ko, 2024) improves routing by utilizing two loss functions one to avoid poor paths and another to optimize the best paths for expert specialization but still fails to address the fundamental issue in spatio-temporal predictions. In tasks with high spatio-temporal heterogeneity, such as traffic flow prediction, the MoE model's reliance on independent expert structures increases inductive bias and reduces overall model performance. Experts with identical structures within MoE introduce a strong inductive bias(Dryden & Hoefler, 2022), further limiting the model's flexibility and adaptability. Additionally, when models with entirely different structures are involved, MoE struggles

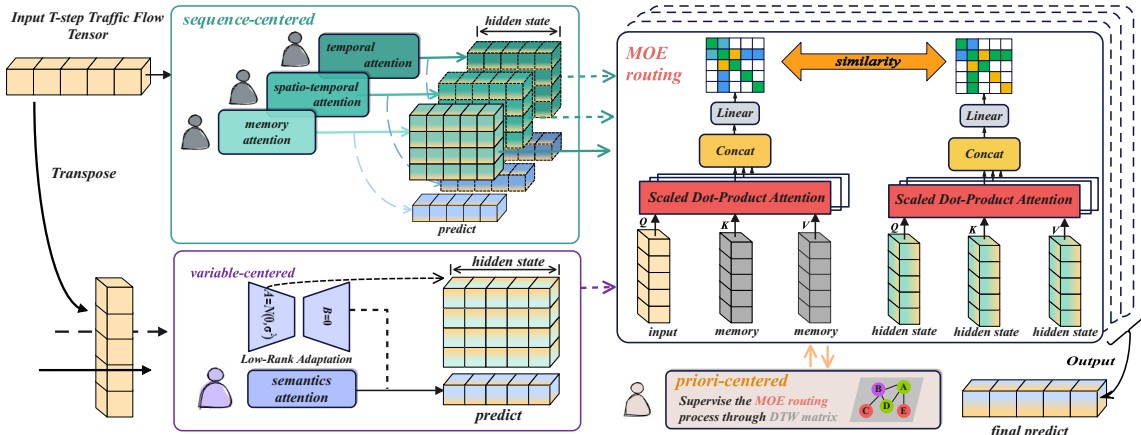

Figure 1: TITAN Overview: Different color schemes represent three different modeling approaches: sequence-centric, variable-centric, and prior knowledge-centric.

to learn the relationship between inputs and outputs, making it challenging to apply the model effectively across diverse tasks. This highlights the requirement for novel approaches that can balance routing precision and expert specialization, particularly in dynamic environments where unexpected events are pivotal.

# 3 METHODS

## 3.1 PROBLEM DEFINITION

**Traffic flow prediction** is a spatio-temporal multivariate time series forecasting problem. Given historical observations $X = \{X_t \in \mathbb{R}^{N \times F}\}$, where $N$ is the number of spatial vertices and $F$ is the number of raw input features (e.g., speed, flow), each time step $t$ is represented by a spatio-temporal graph $G_t = (V, E_t, A_t)$. The task is to predict $T$ future graph signals based on $T'$ historical signals. The model learns a mapping function $f(\cdot) : \mathbb{R}^{T' \times N \times C} \to \mathbb{R}^{T \times N \times C}$, where $C$ represents features derived from processing the original $F$.

## 3.2 MODEL ARCHITECTURE

Although sequence-centric models have shown success in spatio-temporal forecasting, they often struggle to capture complex variable interactions and are challenging to train, leading to reduced accuracy. Additionally, in MoE models, improper memory initialization during pre-training can result in suboptimal routing.TITAN overcomes these limitations by incorporating variable-centric and prior knowledge-centric approaches alongside the traditional sequence-centric method. As shown in Figure 1, TITAN integrates these five experts: 1) three sequence-centric experts (handling sequence-center dependencies in 3.2.1), 2) a variable-centric expert (focusing on variable-center interactions in 3.2.2), and 3) a prior knowledge expert (guiding early-stage routing in 3.2.3). The experts, except for the prior knowledge expert, are based on a lightly modified Transformer architecture, reducing training complexity. The final output is managed through a routing mechanism, ensuring the adaptive selection of experts.

### 3.2.1 Sequence-Centric Modeling

Sequence-centric modeling is a classical approach for spatio-temporal tasks. Building on the backbone of the transformer architecture and incorporating the MOE structure, we have designed three distinct modeling approaches: (1) Temporal Attention expert, (2) Spatio-Temporal Attention expert, and (3) Memory Attention expert. To facilitate explanation, we define the classical Multi-Head Self-Attention (MSA) computation process (Vaswani et al., 2017) as follows: Given the input $X_{data} \in \mathbb{R}^{N \times F}$, the attention result is computed as $X_{out} = \text{MSA}(X_{data}, X_{data}, X_{data})$. The formula for MSA is as follows:

$$Q = X_{in}W_Q, \quad K = X_{in}W_K, \quad V = X_{in}W_V,$$

$$MSA(X_{in}, X_{in}, X_{in}) = \text{softmax}\left(\frac{X_{in}W_Q(X_{in}W_K)^T}{\sqrt{d}}\right) X_{in}W_V, \tag{1}$$

where $W_Q, W_K, W_V \in \mathbb{R}^{F \times d}$ are learnable weight matrices, and $d$ is the dimension of each attention head. At the same time, during the training process, we define that all experts accept the same input $X_{data}$

**Temporal Attention expert** is designed to capture dependencies between time steps, and therefore, it does not involve spatial modeling. However, urban traffic flow is significantly influenced by people's travel patterns and lifestyles, exhibiting clear periodicity (Jiang et al., 2024), such as during morning and evening rush hours. To enhance the ability of temporal attention to capture these periodic patterns, we introduce additional periodic embeddings. Let $w(t)$ be a function that transforms time $t$ into a week index (ranging from 1 to 7). The periodic time embedding $X_w \in \mathbb{R}^{T \times d}$ is obtained by concatenating the embeddings of all $T$ time slices. Finally, the periodic embedding output is calculated by simply adding the data embedding and the periodic embedding:

$$X_{emb} = X_{data} + X_w \quad and \quad X_{ta}, H_{ta} = MSA(X_{emb}, X_{emb}, X_{emb}) \tag{2}$$

By incorporating these periodic embeddings, the temporal attention mechanism is better equipped to capture the cyclical patterns present in traffic data.

**Spatio-Temporal Attention expert** aims to enhance temporal predictions by leveraging the similarity between nodes. Unlike Temporal Attention, it does not use additional periodic embeddings. Instead, it directly applies a linear transformation to map the input $X_{data}$ into a higher-dimensional space. After embedding, the transformed data $X_{emb} \in \mathbb{R}^{T \times N \times C}$ represents the information for $T$ historical time steps, where $N$ is the number of spatial nodes and $C$ is the feature dimension. The process first applies a MSA layer across the $N$ nodes, followed by another MSA layer across the $T$ time steps to capture both spatial and temporal dependencies.

$$\begin{cases} X_{sta}^{(1)}, \_ = MSA(X_{emb}.transpose(1,0,2)) \\ X_{sta}, H_{sta} = MSA(X_{sta}^{(1)}.transpose(1,0,2)) \end{cases} \tag{3}$$

$X_{sta}$ is the output of Spatio-Temporal Attention expert.

**Memory attention expert** is inspired by memory-augmented graph learning (Hong et al.; Lee et al., 2022) and leverages the unique advantages of the memory module in the MOE model. The memory module $M$ consists of a set of vectors with the shape $\mathbb{R}^{N \times m}$, where $m$ represents the memory size. A hypernetwork (Ha et al., 2016) is employed to generate node embeddings conditioned on $M$. Once the new node embeddings are generated, Memory Attention Expert first applies a Graph Convolutional Network (GCN) for spatial feature aggregation, followed by a MSA layer for final prediction. Since the memory module $M$ is supervised by the prior knowledge expert, the Memory Attention Expert is also supervised, allowing it to partially leverage prior knowledge. The process of generating node embeddings and performing GCN is as follows:

$$E = \mathbf{M}W_E, \quad \tilde{A} = softmax(ReLU(EE^\top)), \quad X_{emb} = \tilde{A}X_{data}W_{GCN} \tag{4}$$

where $W_E$ and $W_{GCN}$ are learnable weight matrices. After completing the GCN operation, the resulting node features $H$ are passed to the MSA layer to obtain the final prediction from the Memory Attention Expert:

$$X_{mem}, H_{mem} = MSA(X_{emb}, X_{emb}, X_{emb}) \tag{5}$$

This sequence of operations allows the Memory Attention Expert to capture both spatial and temporal dependencies while also utilizing prior knowledge through the memory module.

### 3.2.2 VARIABLE-CENTERED

In sequence-centric modeling approaches, each time step embedding integrates multiple variables representing latent delayed events or different physical measurements. However, this may fail to learn variable-centric representations and could lead to meaningless attention maps. Additionally, all sequence-centric models share the same inductive bias, which limits the performance of the MOE model. To address these issues, we propose a variable-centric modeling approach. Specifically, time points in each time series are embedded into variable-specific tokens, and the attention mechanism leverages these tokens to capture correlations between different variables. Simultaneously, a feed-forward network is applied to each variable token to learn its nonlinear representation. Unlike sequence-centric models, which generate hidden states based on time steps, the variable-centric model does not share the same hidden state structure, making it difficult to be controlled by the MOE routing mechanism. To resolve this issue, we inject trainable rank decomposition matrices into each layer of the variable-centric model to generate hidden states similar to those in sequence-centric models, thereby reducing inductive bias. For input data $X_{\text{data}} \in \mathbb{R}^{T \times N \times F}$ over $T$ historical time steps, we first map the time steps to a higher-dimensional space through a linear embedding. Specifically, the time steps $T$ are transposed and embedded into the high-dimensional space:

$$X_{\text{emb}} = Embedding(X_{data}.transpose(1, 2, 0)) \tag{6}$$

Here, Embedding is a linear mapping that projects the time steps $T$ into a higher-dimensional space. The expert's output is the result of passing this embedding through the MSA layer, and trainable low-rank matrices are injected between the MSA layers to generate the expert's hidden states

$$X_{enc} = MSA(X_{emb}, X_{emb}, X_{emb}); H_{hidden} = LowRank(X_{enc}) \tag{7}$$

By introducing these trainable low-rank matrices(See appendix A.1 for the detailed introduction), the variable-centric model is able to generate hidden states similar to those of the sequence-centric model, which helps reduce inductive bias and improves the performance of the model within the MOE framework.

### 3.2.3 ANNEALING ROUTING

In regression problems, traditional MoE models tend to make nearly static routing decisions after initialization because the gating network is not effectively guided by gradients from the regression task, as highlighted by Dryden & Hoefler (2022). In this case, the gating network leads to a "mismatch," resulting in routing information that lacks richness and dynamics. Therefore, in time series tasks, the goal of the MoE architecture is typically to learn a direct relationship between input signals and output representations (Lee & Ko, 2024).

For the input $X_{input}$, which includes the input for multiple experts, the memory query process is defined as follows, and the hidden states output by all the experts are further used to calculate $\hat{O}$:

$$O = softmax\left(\frac{X_{out}M^{\top}}{\sqrt{d}}\right)M; \hat{O} = softmax\left(\frac{H_{expert}H_{expert}^{\top}}{\sqrt{d}}\right)H_{expert} \tag{8}$$

where $M$ represents the memory items, a set of learnable vectors. The attention score $O$ is calculated based on $X_{out}$ and $M$, and then normalized using softmax. $H_{expert}$ represents the hidden states of all experts, and $\hat{O}$ is the attention score between the experts' hidden states and the memory items $M$.

Finally, the routing probability for each expert, $p$, is defined as $p = sim(O, \hat{O})$ where $sim(\cdot)$ is a similarity function, such as dot product or cosine similarity, used to measure the similarity between $O$ and $\hat{O}$.

However, this method of calculating the MoE routing probability introduces a problem: during the initial training stages, if $M$ is not properly initialized, it may lead to incorrect routing decisions. These incorrect decisions could guide the experts to train improperly, thus affecting the overall model performance. To address this issue, we improve the routing mechanism by introducing a prior knowledge supervision strategy, ensuring that the routing process is accurately guided during the early stages of model training. This helps avoid unnecessary bias and improves overall model performance.

DTW, or Dynamic Time Warping (Jiang et al., 2024), is a technique widely used in traffic flow forecasting and serves as a reasonable source of prior knowledge (Jin et al., 2024). It captures the similarity and correlation between different nodes. We define the DTW distance between two nodes $i$ and $j$ as: $L_{i,j} = \text{DTW}(X_i, X_j)$, where $X_i$ and $X_j$ represent the time series data of nodes $i$ and $j$, respectively. $L_{i,j}$ denotes the temporal distance or difference between these two nodes. In urban traffic flow forecasting, the relationships between nodes are often sparse, so it is necessary to specify the distances between these nodes. Specifically, we introduce a thresholded Gaussian kernel to adjust the weight matrix between nodes. To reduce noise and avoid treating distant nodes as correlated, we set a threshold $\kappa$ so that when the DTW distance exceeds this threshold, the weight between two nodes is set to 0. The weight matrix $W_{DTW}$ is computed as:

$$W_{i,j} = \begin{cases} \exp\left(-\frac{L_{i,j}^2}{\sigma^2}\right), & \text{if } L_{i,j} \leq \kappa, \\ 0, & \text{if } L_{i,j} > \kappa, \end{cases} \tag{9}$$

where $\sigma$ is the standard deviation of the DTW distances between all nodes, and $\kappa$ is the predefined distance threshold. This approach smooths the distances between nodes using a Gaussian kernel while controlling the sparsity of the relationships based on the distance threshold.

Although matrices constructed based on DTW distances have been proven effective in previous studies, they are not always entirely accurate. To prevent the DTW matrix from overly influencing the training process during the later stages, we were inspired by the cosine annealing strategy for learning rates and designed a prior knowledge cosine annealing strategy. Specifically, during the first $T_{\text{warm}}$ steps of training, we define the routing probability for each expert as: $p = sim(OW_{DTW}, \hat{O})$, where $W_{DTW}$ represents prior knowledge derived from the DTW matrix, analogous to the cosine annealing strategy. During this period, the learning rate is calculated as:

$$lrate = \begin{cases} lr_{min} + (lr_{max} - lr_{min}) \cdot \dfrac{T_{cur}}{T_{warm}}, & \text{for the first } T_{\text{warm}} \text{ steps} \\ lr_{min} + \dfrac{1}{2}(lr_{max} - lr_{min})\left(1 + \cos\left(\dfrac{T_{cur}}{T_{freq}}\pi\right)\right), & \text{otherwise} \end{cases} \tag{10}$$

During the initial stages of training, we introduce $W_{DTW}$ to adjust the routing probability, ensuring a reasonable starting point for the MoE model's routing process. In the later stages of training, we stop multiplying by $W_{DTW}$, thereby preventing incorrect prior knowledge from interfering with the model. This approach allows the model to fully utilize prior knowledge during the early stages while avoiding its potential negative effects in the later stages, thus improving overall model performance.

## 4 EXPERIMENT

In this section, we conduct experiments on two benchmark datasets: METR-LA and PEMS-BAY. METR-LA and PEMS-BAY contain four months of speed data recorded by 207 sensors on Los Angeles freeways and 325 sensors in the Bay Area, respectively(Li & Shahabi, 2018). Before training TITAN, we have performed

z-score normalization. In the case of METR-LA and PEMS-BAY, we use 70% of the data for training, 10% for validation, and 20% for evaluation.

Table 1: The experimental results are based on 14 baseline models and TITAN across three real-world datasets. **Bold** indicates the best performance, and underline indicates the second-best performance. "Promotion" represents the improvement margin of TITAN compared to the best baseline model.

| METR-LA | 15 min | | | 30 min | | | 60 min | | |
|---|---|---|---|---|---|---|---|---|---|
| | MAE | RMSE | MAPE | MAE | RMSE | MAPE | MAE | RMSE | MAPE |
| STGCN (2018) | 2.88 | 5.74 | 7.62% | 3.47 | 7.24 | 9.57% | 4.59 | 9.40 | 12.70% |
| DCRNN (2018) | 2.77 | 5.38 | 7.30% | 3.15 | 6.45 | 8.80% | 3.60 | 7.59 | 10.50% |
| Graph-WaveNet (2019) | 2.69 | 5.15 | 6.90% | 3.07 | 6.22 | 8.37% | 3.53 | 7.37 | 10.01% |
| GMAN (2020) | 2.80 | 5.55 | 7.41% | 3.12 | 6.49 | 8.73% | 3.44 | 7.35 | 10.07% |
| MTGNN (2020) | 2.69 | 5.18 | 6.86% | 3.05 | 6.17 | 8.19% | 3.49 | 7.23 | 9.87% |
| StemGNN (2020) | 2.56 | 5.06 | 6.46% | 3.01 | 6.03 | 8.23% | 3.43 | 7.23 | 9.85% |
| AGCRN (2020) | 2.86 | 5.55 | 7.55% | 3.25 | 6.57 | 8.99% | 3.68 | 7.56 | 10.46% |
| CCRNN (2021) | 2.85 | 5.54 | 7.50% | 3.24 | 6.54 | 8.90% | 3.73 | 7.65 | 10.59% |
| PM-MemNet (2022) | 2.65 | 5.29 | 7.01% | 3.03 | 6.29 | 8.42% | 3.46 | 7.29 | 9.97% |
| MegaCRN (2023) | 2.52 | 4.94 | 6.44% | 2.93 | 6.06 | 7.96% | 3.38 | 7.23 | 9.72% |
| TESTAM (2024) | 2.54 | 4.93 | 6.42% | 2.96 | 6.04 | 7.92% | 3.36 | 7.09 | 9.67% |
| RGDAN (2024) | 2.69 | 5.20 | 7.14% | 2.99 | 5.98 | 8.07% | 3.26 | 7.02 | 9.73% |
| AdpSTGCN (2024) | 2.59 | 5.11 | 6.68% | 2.96 | 6.08 | 8.02% | 3.40 | 7.21 | 9.45% |
| STD-MAE (2024) | 2.62 | 5.02 | 6.70% | 2.99 | 6.07 | 8.04% | 3.40 | 7.07 | 9.59% |
| TITAN | **2.41** | **4.43** | **5.85%** | **2.72** | **5.33** | **7.07%** | **3.08** | **6.21** | **8.43%** |
| Promotion | +4.37% | +10.14% | +8.88% | +7.17% | +10.87% | +10.73% | +5.52% | +11.53% | +10.79% |

| PEMS-BAY | 15 min | | | 30 min | | | 60 min | | |
|---|---|---|---|---|---|---|---|---|---|
| | MAE | RMSE | MAPE | MAE | RMSE | MAPE | MAE | RMSE | MAPE |
| STGCN(2018) | 1.36 | 2.96 | 2.90% | 1.81 | 4.27 | 4.17% | 2.49 | 5.69 | 5.79% |
| DCRNN(2018) | 1.38 | 2.95 | 2.90% | 1.74 | 3.97 | 3.90% | 2.07 | 4.74 | 4.90% |
| Graph-WaveNet(2019) | 1.3 | 2.74 | 2.73% | 1.63 | 3.7 | 3.67% | 1.95 | 4.52 | 4.63% |
| GMAN(2020) | 1.35 | 2.9 | 2.87% | 1.65 | 3.82 | 3.74% | 1.92 | 4.49 | 4.52% |
| MTGNN(2020) | 1.32 | 2.79 | 2.77% | 1.65 | 3.74 | 3.69% | 1.94 | 4.49 | 4.53% |
| StemGNN(2020) | 1.23 | 2.48 | 2.63% | N/A | N/A | N/A | N/A | N/A | N/A |
| AGCRN(2020) | 1.36 | 2.88 | 2.93% | 1.69 | 3.87 | 3.86% | 1.98 | 4.59 | 4.63% |
| CCRNN(2021) | 1.38 | 2.9 | 2.90% | 1.74 | 3.87 | 3.90% | 2.07 | 4.65 | 4.87% |
| PM-MemNet(2022) | 1.34 | 2.82 | 2.81% | 1.65 | 3.76 | 3.71% | 1.95 | 4.49 | 4.54% |
| MegaCRN(2023) | 1.28 | 2.72 | 2.67% | 1.6 | 3.68 | 3.57% | 1.88 | 4.42 | 4.41% |
| TESTAM(2024) | 1.29 | 2.77 | 2.61% | 1.59 | 3.65 | 3.56% | 1.85 | 4.33 | 4.31% |
| RGDAN(2024) | 1.31 | 2.79 | 2.77% | 1.56 | 3.55 | 3.47% | 1.82 | 4.20 | 4.28% |
| AdpSTGCN(2024) | 1.29 | 2.75 | 2.69% | 1.64 | 3.69 | 3.67% | 1.92 | 4.49 | 4.62% |
| STD-MAE (2024) | 1.23 | 2.62 | 2.56% | 1.53 | 3.53 | 3.42% | 1.77 | 4.20 | 4.17% |
| TITAN | **1.12** | **2.21** | **2.29%** | **1.46** | **3.24** | **3.15%** | **1.69** | **3.79** | **3.85%** |
| Promotion | +8.94% | +10.89% | +10.55% | +4.56% | +8.21% | +7.89% | +4.51% | +9.76% | +7.67% |

## 4.1 EXPERIMENTAL SETTINGS

For all two datasets, we use Xavier initialization to initialize the parameters and embeddings. Hyperparameters are obtained via a limited grid search on the validation set with hidden size = {16, 32, 48, 64, 80}, memory size = {16, 32, 48, 64, 80}, and layers = {1, 2, 3, 4, 5}. Figure 2 illustrates the results of the hyperparameter search and the corresponding model parameter counts. Different colored lines represent various error metrics, while the bar chart shows the model's parameter count under different hyperparameters. We use the Adam optimizer with $\beta_1 = 0.9$, $\beta_2 = 0.98$, and $\epsilon = 10^{-9}$. The learning rate is set to $lr_{\max} = 3 \times 10^{-3}$, with the scheduling method as described in Section 3.2.3. We follow the traditional 12-sequence (1-hour) input and 12-sequence output prediction setting for METR-LA and PEMS-BAY. Mean absolute error (MAE) is used as the loss function, while root mean square error (RMSE) and mean absolute percentage error (MAPE) are used as evaluation metrics. All experiments are conducted using 8 RTX A6000 GPU

We compare TITAN with 14 baseline models: (1) STGCN (Yu et al., 2018), a model containing GCN and CNN; (2) DCRNN (Li et al., 2018), a model with graph convolutional recurrent units; (3) Graph-WaveNet (Wu et al., 2019) with parameterized adjacency matrix; (4) GMAN (Zheng et al., 2020), an attention-based model; (5) MTGNN (Wu et al., 2020c); (6) StemGNN (Cao et al., 2021); (7) AGCRN (Bai et al., 2020), advanced models with adaptive matrices; (8) CCRNN (Ye et al., 2021); (9) MAF-GNN (Xu et al., 2021), models with multiple adaptive matrices; (10) PM-MemNet (Lee et al., 2022); (11) MegaCRN (Jiang et al., 2023a), with memory units; (12) TESTAM (Lee & Ko, 2024), a model that also uses the MOE structure and integrates different graph structures; (13) AdpSTGCN (Zhang et al., 2024), a model based on adaptive graph convolution; (14) STD-MAE (Gao et al., 2024), a state-of-the-art model employing two decoupled masked autoencoders to reconstruct spatiotemporal series.

## 4.2 EXPERIMENTAL RESULTS

The experimental results are shown in Table 4. TITAN outperforms all other models, demonstrating an average improvement of approximately 9% across all forecasting horizons compared to the best baseline. It is noteworthy that we compare our reported results from respective papers with those obtained by replicating the official code provided by the authors. Sequence-centric modeling approaches, including both static graph models (DCRNN, RGDAN, MTGNN, CCRNN) and dynamic graph models (GMAN, AdpSTGCN), exhibit competitive performance in capturing spatiotemporal dependencies. However, STD-MAE achieves superior performance by reconstructing time series in both sequential and variational dimensions to capture complex spatiotemporal relationships. TESTAM, employing a MOE structure to simultaneously capture multiple spatial relationships, shows suboptimal performance. However, it is still subject to the inherent bias shared among homogeneous experts, resulting in performance inferior to TITAN. In contrast, our model TITAN demonstrates superiority over all other models, including those with learnable matrices. The results underscore the critical importance of correctly handling sudden events in traffic flow prediction. A more detailed efficiency analysis is provided in Appendix A.2, and the robustness for each indicator is visualized in Appendix A.5.

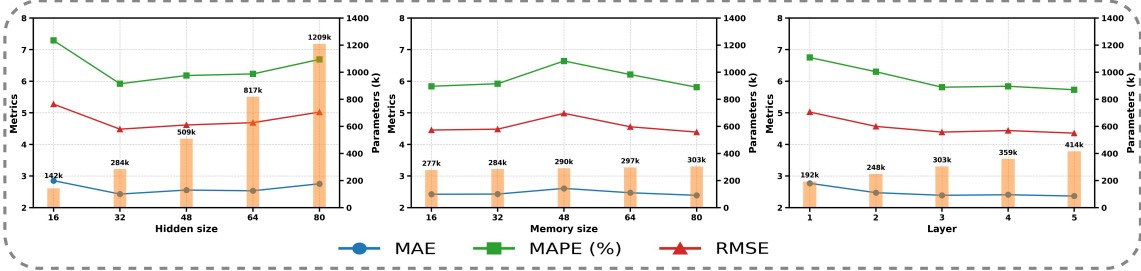

Figure 2: TITAN model evaluation indicators under different parameter settings, marked in different colors

## 4.3 ABLATION STUDY

The ablation study has two goals: to evaluate the actual improvements achieved by each method and to test two hypotheses: (1) For heterogeneous MOE models, organizing the model using a low-rank adaptive approach is beneficial; (2) Organizing the experts through supervised routing is effective. To achieve these goals, we designed a set of TITAN variants as described below:

**w/o Semantics**: This variant removes the heterogeneous semantics expert. In this case, TITAN only retains a sequence-centric modeling approach, which can be seen as similar to the TESTAM structure.

**Ensemble**: The final output is not computed using MoE but as a weighted sum of the outputs of each expert via a gating network. This setup can use all heterogeneous experts but cannot use the a priori expert because there is no routing process.

**w/o priori expert**: This variant directly removes the priori expert, and the routing process is only supervised by the classification loss.

**Replaced priori expert**: This variant changes the organization of the priori expert by adding a temporal prediction module. In this structure, the priori expert can be seen as a sequence-centric modeling approach. The

Table 2: Ablation study results across all prediction windows (i.e., average performance)

| Ablation | METR-LA | | | PEMS-BAY | | |
|---|---|---|---|---|---|---|
| | MAE | RMSE | MAPE | MAE | RMSE | MAPE |
| w/o Semantics | 2.82 | 5.53 | 7.35% | 1.45 | 3.11 | 3.21% |
| Ensemble | 3.66 | 6.08 | 8.13% | 1.59 | 3.69 | 3.33% |
| w/o a priori | 2.82 | 5.46 | 7.42% | 1.43 | 3.10 | 3.17% |
| Replaced a priori | 3.06 | 5.96 | 8.19% | 1.60 | 3.47 | 3.33% |
| **TITAN** | **2.74** | **5.33** | **7.13%** | **1.42** | **3.08** | **3.10%** |

experimental results shown in Table 4.3 indicate that our hypotheses are supported and TITAN is a complete and inseparable system. The results of w/o Semantics show that the heterogeneous semantics expert also improves the predictions. The results of Ensemble demonstrate that the MoE approach significantly improves traffic prediction quality. The results of w/o a priori expert suggest that correctly using prior knowledge to supervise the routing process in MoE can help avoid local optima to a certain extent, thus enhancing the model's performance. The results of Replacing a priori expert show that simply adding an expert to provide prior knowledge is ineffective, as the MoE model struggles to directly assess the role of prior knowledge. Other elaborate results are provided in the Appendix A.3.For the role that each expert plays in the training process we provide SOME MEANINGFUL OBSERVATIONS in Appendix A.4

## 5 CONCLUSION

In this paper, we propose a heterogeneous mixture of expert model (TITAN) that can simultaneously perform sequence-centric, variable-centric, and prior knowledge-centric modeling, incorporating cosine annealing of prior knowledge during the training process. TITAN achieves state-of-the-art performance on two real-world datasets, delivering an average improvement of approximately 9% compared to the previous best baseline. Through ablation experiments, we demonstrated the effectiveness of each module within TITAN. Additionally, we conducted parameter and efficiency analyses to further assess TITAN's performance. For future work, we aim to introduce more intelligent algorithms, such as heuristic methods, to optimize the routing process in TITAN and extend its application to multivariate time series forecasting tasks.

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

## A  APPENDIX

### A.1  DETAILED INTRODUCTION TO LOW-RANK MATRIX ADAPTATION

---
**Algorithm 1** Low-Rank Adaptation for Semantic expert
---
1: **Input:**
2:    Initial model parameters $\Theta$, low-rank matrices $W_A \in \mathbb{R}^{d_{\mathrm{model}} \times r}, W_B \in \mathbb{R}^{r \times hidden\_size}$
3:    Input data $X_{enc}$
4:    Target data $Y$
5:    Training hyperparameters: learning rate $\eta$, rank $r$, number of epochs $N_{\mathrm{epoch}}$
6: **Output:**
7:    model parameters affected by optimized low-rank matrix $\Theta^*$
8:    Optimized low-rank matrices $W_A^*, W_B^*$
9: Initialize model parameters $\Theta_0$, low-rank matrices $W_A, W_B$
10: **repeat**
11:     **Step 1: Generate embeddings and forecasts**
12:         Compute embeddings $enc\_out \leftarrow P_\Theta(X_{enc})$
13:     **Step 2: Compute prediction loss**
14:         $L_P(\hat{Y}, Y)$
15:     **Step 3: Update model parameters**
16:         $\Theta \leftarrow \Theta - \eta \cdot \nabla_\Theta L_P$
17:     **Step 4: Apply low-rank adaptation**
18:         $Z \leftarrow W_A \cdot enc\_out$    (low-rank projection)
19:         $\hat{Y}_{new} \leftarrow W_B \cdot Z$    (dimension recovery)
20:     **Step 5: Compute loss after low-rank transformation**
21:         $L_G(\hat{Y}_{new}, Y)$
22:     **Step 6: Update low-rank matrices $W_A$ and $W_B$**
23:         $W_A \leftarrow W_A - \eta \cdot \nabla_{W_A} L_G$
24:         $W_B \leftarrow W_B - \eta \cdot \nabla_{W_B} L_G$
25:     **Step 7: Check for convergence**
26: **until** maximum epochs $N_{\mathrm{epoch}}$ reached or convergence condition is satisfied
27: **Return:** model parameters affected by optimized low-rank matrix $\Theta^*$ and low-rank matrices $W_A^*, W_B^*$

---

The pseudocode above outlines the process of Low-Rank Adaptation in the Semantic Expert of the MoE model. The low-rank matrices $W_A$ and $W_B$ act as a bridge between the hidden layers and the final output of the Semantic Expert. In **Step 4**, a low-rank projection is applied to the embeddings using $W_A$, followed by dimensional recovery using $W_B$. The model iteratively updates both the main model parameters $\Theta$ and the low-rank matrices $W_A$ and $W_B$, ensuring that the low-rank matrices adapt to the data alongside the model's learning process.

## A.2 EFFICIENCY ANALYSIS

To analyze the computational cost, we use seven baseline models:

**STGCN** (Yu et al., 2018), a lightweight approach that predicts future traffic conditions by utilizing GCN and CNN;

**DCRNN** (Li et al., 2018), a well-established traffic forecasting model that integrates graph convolution into recurrent units, depending on a predefined graph structure;

**Graph-WaveNet** (Wu et al., 2019), which leverages graph embeddings for traffic prediction;

**GMAN** (Zheng et al., 2020), a spatio-temporal attention model designed for traffic forecasting;

**MegaCRN** (Jiang et al., 2023b), a state-of-the-art model that uses GCRNN combined with memory networks;

**TESTAM** (Lee & Ko, 2024), a MoE-based model for traffic prediction that routes experts through a pseudo-label classification task;

**RGDAN** (Fan et al., 2024), an advanced approach that utilizes random graph attention, noted for its computational efficiency.

We also examined other models for comparison, but after careful consideration, we chose to exclude them. For example, MTGNN and StemGNN were excluded as they are improved versions of Graph-WaveNet, offering similar computational costs. Likewise, AGCRN and CCRNN were left out as they are DCRNN variants with minimal changes in computational cost. While PM-MemNet and MegaCRN use sequence-to-sequence modeling with shared memory units, PM-MemNet suffers from computational bottlenecks due to its stacked memory units, requiring L times the computational resources of MegaCRN. All models were tested following TESTAM's experimental setup, and both training and inference times were recorded using the same hardware.

Table 3: Comparison of computational costs and performance improvements across different baseline models for traffic prediction tasks.

| Model | Training time/epoch | Inference time | params |
|---|---|---|---|
| STGCN | 14.8 secs | 16.70 secs | 320k |
| DCRNN | 122.22 secs | 13.44 secs | 372k |
| Graph-WaveNet | 48.07 secs | 3.69 secs | 309k |
| GMAN | 312.1 secs | 33.7 secs | 901k |
| TESTAM | 150 secs | 7.96 secs | 224k |
| MegaCRN | 84.7 secs | 11.76 secs | 339k |
| RGDAN | 68 secs | 6.3 secs | 337k |
| TITAN | 160 secs | 6.6 secs | 283k |

As shown in Table 3, Even though TITAN employs five individual experts, we highlight that the prior knowledge-centric expert does not contribute to the prediction process. Additionally, each expert in the system has fewer layers, which results in a lower parameter count compared to other models, thus significantly reducing the overall computational load. As a result, TITAN's additional computational cost remains manageable. Compared to TESTAM, TITAN incurs an extra 10 seconds of computation time but delivers around a 10% improvement in performance, making the additional cost a worthwhile investment.

## A.3 DETAILED ABLATION STUDY RESULTS

Table 4: Detailed Ablation Study Results

| METR-LA | 15 min | | | 30 min | | | 60 min | | |
|---|---|---|---|---|---|---|---|---|---|
| | **MAE** | **RMSE** | **MAPE** | **MAE** | **RMSE** | **MAPE** | **MAE** | **RMSE** | **MAPE** |
| w/o Semantics | 2.43 | 4.50 | 5.89% | 2.81 | 5.51 | 7.32% | 3.21 | 6.60 | 8.84% |
| w/o a priori | 2.59 | 4.72 | 6.30% | 2.76 | 5.38 | 7.38% | 3.13 | 6.29 | 8.59% |
| Replaced a priori | 2.49 | 4.63 | 6.17% | 2.79 | 5.44 | 7.61% | 3.90 | 7.83 | 10.80% |
| Ensemble | 3.14 | 6.79 | 7.60% | 3.60 | 8.16 | 8.89% | 4.26 | 9.44 | 11.14% |
| **TITAN** | **2.41** | **4.43** | **5.85%** | **2.72** | **5.33** | **7.07%** | **3.08** | **6.24** | **8.47%** |
| **PEMS-BAY** | **15 min** | | | **30 min** | | | **60 min** | | |
| | **MAE** | **RMSE** | **MAPE** | **MAE** | **RMSE** | **MAPE** | **MAE** | **RMSE** | **MAPE** |
| w/o Semantics | 1.14 | 2.22 | 2.36% | 1.48 | 3.27 | 3.26% | 1.72 | 3.85 | 4.01% |
| w/o a priori | 1.17 | 2.41 | 2.36% | 1.42 | 3.00 | 3.26% | 1.69 | 3.90 | 3.90% |
| Replaced a priori | 1.24 | 2.63 | 2.49% | 1.52 | 3.33 | 3.17% | 1.96 | 4.46 | 4.33% |
| Ensemble | 1.24 | 2.61 | 2.46% | 1.53 | 3.65 | 3.19% | 2.00 | 4.81 | 4.36% |
| **TITAN** | **1.12** | **2.21** | **2.29%** | **1.46** | **3.24** | **3.15%** | **1.69** | **3.79** | **3.85%** |

The results of the ablation study, presented in Table 4, strongly support our hypotheses. Below, we provide a more detailed explanation of each ablation variant's outcomes:

**w/o Semantics**: The performance drops when the heterogeneous Semantics expert is removed, indicating that the use of heterogeneous experts significantly improves predictive accuracy. Without the semantics expert, the model behaves similarly to TESTAM, demonstrating that relying solely on sequence-centric modeling limits the model's ability to capture complex relationships between variables.

**Ensemble**: The results for this variant reveal that the MoE approach, which dynamically selects experts, outperforms a simple ensemble approach where expert outputs are combined through a weighted sum. The lack of routine supervision in the ensemble setup prevents optimal expert specialization, leading to lower performance. Furthermore, the absence of the a priori expert in this setup shows the importance of including prior knowledge in the routing process.

**w/o a priori expert**: The performance decrease in this variant highlights the importance of the a priori expert in guiding the routing process. Without prior knowledge to supervise the routing, the model is more prone to suboptimal routing decisions, potentially leading to local minima during training. This supports the idea that incorporating prior knowledge in MoE helps avoid these pitfalls and enhances overall model performance.

**Replaced a priori expert**: Replacing the a priori expert with a sequence-centric prediction module results in lower performance, demonstrating that the mere inclusion of an additional expert is not sufficient to capture prior knowledge effectively. This variant confirms that the MoE model struggles to leverage the added temporal module and underscores the importance of properly utilizing prior knowledge to guide the routing process. Simply adding another expert without a clear distinction in function does not provide the desired performance gains.

By analyzing these detailed ablation results, we can conclude that TITAN is an integrated and cohesive model where each component plays a crucial role. Both the low-rank adaptive organization and supervised routing are essential for achieving state-of-the-art performance in traffic prediction.

## A.4 SOME MEANINGFUL OBSERVATIONS

In this section, we provide a deeper analysis of the expert selection behavior as presented in Table 5, which illustrates how the number of times each expert is selected changes with the prediction horizon (i.e., 3-step, 6-step, and 12-step predictions) on the PEMS-BAY dataset.

Table 5: The number of times each expert is selected changes with the length of time

| **PEMS-BAY** | **3step** | **6step** | **12step** |
|---|---|---|---|
| Temporal Attention expert | 13 | 283 | 11469918 |
| Spatio-Temporal Attention expert | **9235854** | 265436 | **23445344** |
| Memory attention expert | 769 | 671936 | 19156 |
| Semantics-centered expert | 871189 | **19276045** | 5481282 |

The Temporal Attention expertis rarely selected for short-term predictions (3-step horizon), with only 13 selections, suggesting that short-term forecasts do not heavily depend on temporal dynamics alone. However, its selection increases moderately to 283 at the 6-step horizon, indicating growing importance as the prediction length increases. By the 12-step horizon, its selection count dramatically rises to 11,469,918, highlighting the critical role of temporal dependencies in long-term predictions. This indicates that the Temporal Attention expert becomes indispensable for longer-term forecasting, likely due to the increasing need to capture sequential patterns over time. The Semantics-Centered expert is most frequently selected during mid-term predictions (6-step horizon) and is utilized across all time steps, demonstrating the effectiveness of our low-rank adaptive method in designing this expert. The consistent selection of the Semantics expert across different horizons shows its robust contribution to the model's overall predictive capabilities.

## A.5 FURTHER VISUAL CONCLUSIONS

In this section, we provide a more detailed explanation of the results discussed in the main text, with additional insights into TITAN's performance across various datasets and time steps. Specifically, we have generated box plots illustrating the error distributions for each evaluation metric, segmented by dataset and prediction time step, as shown in Figure 3.

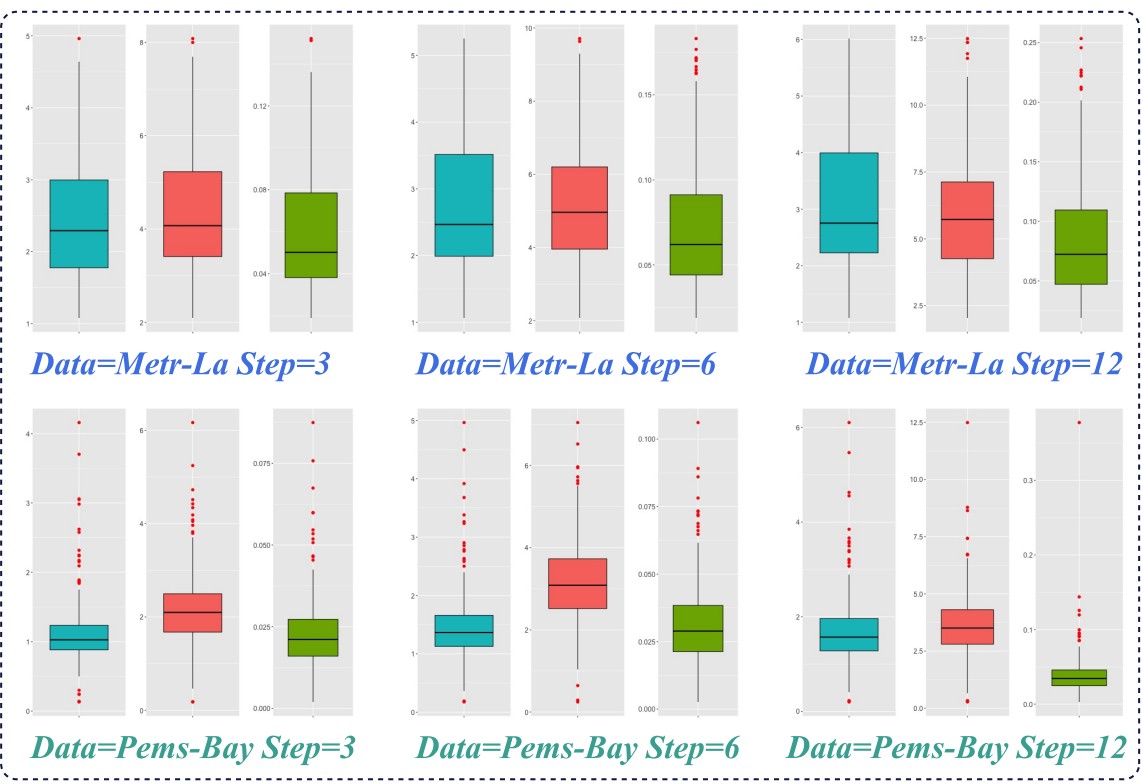

Figure 3: Box plots illustrating the error distributions of TITAN across all nodes for the Metr-LA and Pems-Bay datasets, segmented by prediction time steps.

**Performance Stability across Datasets**: As observed in Figure 3, TITAN demonstrates a more stable performance on the Pems-Bay dataset compared to Metr-LA. This can be attributed to the inherent characteristics of the datasets: Pems-Bay, having a denser and more consistent traffic flow, enables TITAN to better capture the spatial and temporal dependencies across different nodes, resulting in reduced variability in prediction errors. In contrast, the Metr-LA dataset, which represents a more dynamic urban traffic environment with irregular patterns, introduces greater variability in traffic conditions, leading to slightly more fluctuation in TITAN's performance. Nonetheless, even with the increased complexity in Metr-LA, TITAN manages to maintain strong predictive capabilities, as evidenced by the box plots' overall narrow range of error distributions.

**Adaptability to Prediction Time Steps**: One of TITAN's key strengths, as reflected in the results, is its robustness across varying prediction horizons. As shown in Figure 3, TITAN's error distributions remain relatively stable as the prediction step increases across both datasets. This consistency indicates that TITAN is highly adaptable to different prediction time steps, whether forecasting short-term traffic conditions (i.e., 15 minutes ahead) or long-term trends (i.e., 60 minutes ahead). Unlike some models that suffer from significant performance degradation as the prediction window widens, TITAN's architecture, which integrates both sequence-centric and variable-centric experts, allows it to maintain accuracy over varying time scales. Furthermore, the lack of significant error spikes as the prediction step increases highlights TITAN's ability to generalize well to different forecasting tasks, suggesting that its mixture of expert structure enables it to capture both short-term fluctuations and long-term patterns effectively.

## A.6 REPRODUCIBILITY STATEMENT

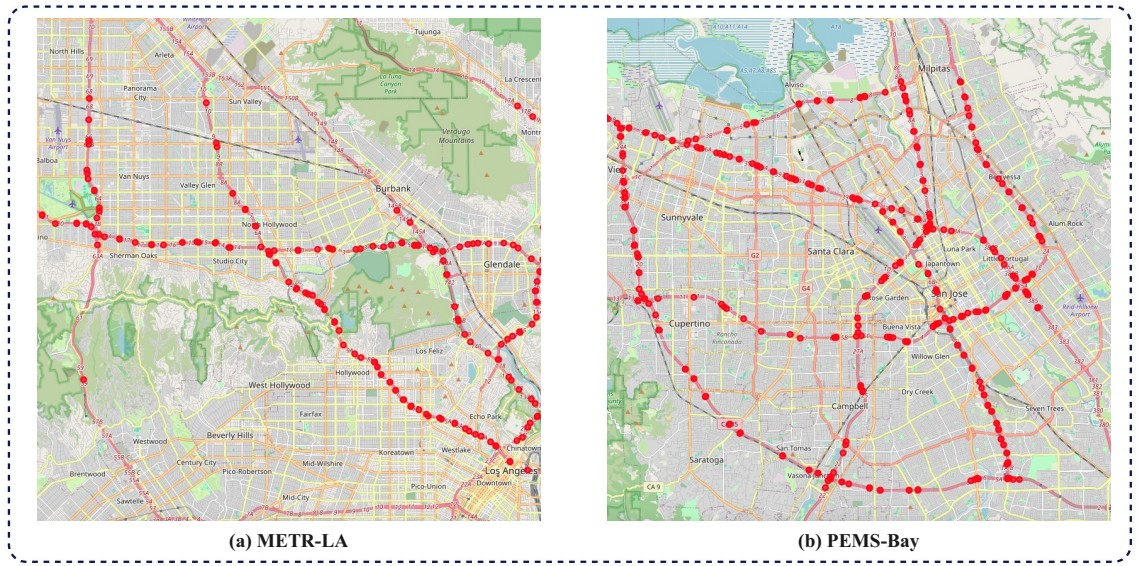

| (a) METR-LA | (b) PEMS-Bay |

Figure 4: Sensor distribution of the (a)METR-LA and (b)PEMS-BAY dataset.

We perform experiments using two extensive real-world datasets powered by (Li et al., 2018):

- METR-LA: This dataset provides traffic information gathered from loop detectors installed on high-ways in Los Angeles County(Jagadish et al., 2014). For our experiments, we use data from 207 selected sensors and collect traffic data over a 4-month period from March 1st, 2012, to June 30th, 2012. The total number of traffic data points observed amounts to 6,519,002.

- PEMS-BAY: This traffic dataset is provided by California Transportation Agencies (CalTrans) via the Performance Measurement System (PeMS). We gather data from 325 sensors located in the Bay Area over a 6-month period, spanning from January 1st, 2017, to May 31st, 2017. The total number of observed traffic data points is 16,937,179.

The distribution of sensors for both datasets is shown in Figure 4.

All model parameters and the selection process have been described in detail in the paper. We trained the models using 8 NVIDIA GPUs with a fixed random seed to ensure reproducibility. The weight files generated during training will be made publicly available along with the code after the paper is officially accepted, allowing researchers to access and verify the results.

