# OpenReview forum: "A Time Series is Worth Five Experts: Heterogeneous Mixture of Experts for Traffic Flow Prediction"
_ICLR.cc/2025/Conference — ICLR 2025 Conference Withdrawn Submission_

### Official Review · Reviewer_xfFa · 2024-10-21

**Soundness:** 3
**Presentation:** 3
**Contribution:** 2
**Rating:** 5
**Confidence:** 4

**Summary:**

The paper introduces TITAN, a heterogeneous mixture of expert model designed to improve traffic forecasting. TITAN integrates both sequence-centric and variable-centric modeling techniques alongside a supervised routing mechanism driven by prior knowledge. By leveraging four distinct expert groups and a low-rank adaptation method, the model aims to capture diverse spatio-temporal dependencies in traffic data. Experimental results on two real-world datasets show performance improvements over state-of-the-art models, demonstrating the effectiveness of TITAN.

**Strengths:**

1. This paper studies a critical task, i.e., traffic forecasting, which has wide-ranging applications in smart cities, autonomous driving, and transportation management.
2. The model design intuitively mirrors the real-world interactions in traffic systems, where periodic temporal patterns and cross-node relationships need to be captured for accurate forecasting.
3. The experimental results showcase TITAN’s superior performance across two benchmark datasets (METR-LA and PEMS-BAY), with improvements over state-of-the-art models in three popular metrics across different prediction horizons.

**Weaknesses:**

1. The novelty of the proposed model appears limited. The three types of sequence-centric modeling experts have been extensively studied in previous literature. Furthermore, the idea of variable-centered modeling seems to draw heavily from the iTransformer [1], which has already explored variable-specific tokenization and attention mechanisms.
2. This paper lacks detailed analysis regarding the computational complexity of the proposed model, making it difficult to evaluate the model's efficiency and scalability. Since attention mechanisms typically exhibit quadratic complexity with respect to the number of nodes, this could result in substantial computational overhead when scaling the model to large-scale road networks.
3. The evaluation scope is limited. The paper focuses on only two small datasets (i.e., METR-LA and PEMS-BAY), which may not be representative of broader traffic forecasting tasks. Testing the model on larger datasets, such as those with more complex and extensive urban traffic networks (e.g., LargeST [2]), would provide a better assessment of its real-world applicability.
4. Some important recent studies, such as [3] and [4], are not discussed in the related works.

[1] Liu, Yong, et al. "iTransformer: Inverted Transformers Are Effective for Time Series Forecasting." In The Twelfth International Conference on Learning Representations.

[2] Liu, Xu, et al. "Largest: A benchmark dataset for large-scale traffic forecasting." Advances in Neural Information Processing Systems 36 (2024).

[3] Li, Shuhao, et al. "ST-MoE: Spatio-Temporal Mixture-of-Experts for Debiasing in Traffic Prediction." Proceedings of the 32nd ACM International Conference on Information and Knowledge Management. 2023.

[4] Jiang, Wenzhao, et al. "Interpretable cascading mixture-of-experts for urban traffic congestion prediction." Proceedings of the 30th ACM SIGKDD Conference on Knowledge Discovery and Data Mining. 2024.

**Questions:**

Please see in weaknesses.

---

### Official Review · Reviewer_Bm61 · 2024-10-26

**Soundness:** 3
**Presentation:** 3
**Contribution:** 2
**Rating:** 5
**Confidence:** 4

**Summary:**

This paper propose TITAN framework, which incorporates sequence-centric, variable-centric experts and a leader expert supervising the routing process to capture the complex spatio-temporal dependencies. They have also designed an expert strategy to improve the performance of the memory query process of MoE. Extensive experiments have also been performed on real world datasets.

**Strengths:**

1. This paper presents a novel combination of experts, addressing temporal focus dependency, spatio-temporal relationships, and memory collection strategies. Together, these experts effectively capture the intricate correlations within the data. Additionally, the framework design of TITAN allows for flexibility and adaptability, making it applicable to a wide range of spatio-temporal prediction tasks.
2. The main experiments employ multiple state-of-the-art methods for evaluation, with TITAN outperforming all the compared approaches.
3. The topic about the Mixture of Experts (MoE) framework is worth investigating, as it tackles the challenge of coordinating specialized experts to capture diverse dependencies in complex data and each expert could be adapted to other domains.

**Weaknesses:**

1. The paper introduces a Memory Attention Expert for long-term prediction but lacks an explicit priority mechanism for memory storage. While attention can select important components based on similarity, it doesn’t fully solve long-term prediction issues. If past relevant information is forgotten, the expert may fail to capture long-term patterns effectively.
2. The DTW matrix for prior knowledge may hinder the performance of the sequence-centric expert. DTW assumes static temporal patterns based on historical data, which may conflict with the sequence-centric expert's goal of dynamically learning time dependencies from the input. This reliance on fixed temporal similarities could limit the  ability of the expert to adapt to new time patterns during training.
3. The experiments raise some concerns, as the paper focuses on traffic flow, but the datasets used are for speed, which creates a disconnect between the methods and the topic, making the results less convincing.
4. The experiments should include longer prediction intervals, as the current 15-60 minute range is insufficient to fully evaluate the memory attention expert, which is intended to be more beneficial for long-term prediction tasks.
5. Table 1 contains a typo regarding the number of real-world datasets. It claims to use three, but only two are included in the experiments.
6. No code has been provided, making it difficult to evaluate the methods and reproduce the results.

**Questions:**

1. How can the Memory attention expert select which parts of the memory should be kept?
2. There is confusion regarding the variable-centered experts. From my understanding, these experts should focus on multiple variables like inflow, speed, and demand in a given area. However, the experiment only uses speed data, which raises the question of how variable-centered experts differ from sequence-centered ones. If both experts operate on a single dataset, their impact seems nearly identical.
3. In the annealing routing method, how do you resolve the potential conflict between the DTW method, which assumes static temporal patterns, and the sequence-centric expert, which dynamically captures time dependencies?
4. This paper claims to use a graph-based approach, but there is little mention of graph construction details. For instance, how is the adjacency matrix generated? Is it provided by the dataset or learned during training?

---

### Official Review · Reviewer_EJ4B · 2024-11-02

**Soundness:** 3
**Presentation:** 2
**Contribution:** 2
**Rating:** 5
**Confidence:** 4

**Summary:**

The paper proposes a Heterogeneous Mixture of Experts (MoE) model named TITAN for traffic flow prediction. The model addresses the limitations of existing sequence-centric traffic prediction models by incorporating variable-centric and prior knowledge-centric modeling techniques. TITAN consists of three sequence-centric experts, one variable-centric expert incorporated by low-rank adaptive matrices, and a leader expert to supervise routing decisions. An expert annealing strategy is further employed to gradually reduce supervision from the leader expert during training. Experiments on two public datasets, METR-LA and PEMS-BAY, demonstrate that TITAN outperforms state-of-the-art models in terms of prediction accuracy.

**Strengths:**

1. **Meaningful Research Problem**: Though MoE has demonstrated extraordinary capacity in NLP and CV fields, its application in spatio-temporal (ST) problems is relatively underexplored. Therefore, the paper contributes to the understanding of MoE's potential in ST applications.
2. **Exploring from Important Aspects:** The paper explores heterogeneous expert design and stable routing strategies, which are crucial steps when applying MoE to spatio-temporal applications.
3. **Clarity in Writing**: The paper is fairly well-structured, with a clear explanation of the proposed model and its components, making it accessible for readers to understand the methodology and contributions.

**Weaknesses:**

1. **Unclear Motivation**:

   **a) Missing definitions:** The terms 'sequence-centric' and 'variable-centric' are mentioned multiple times, but their definitions are unclear. This lack of clarity makes it difficult to understand why existing models belong to differing categories and what specific limitations they have. For examples:

   - The paper argues that models like GraphWaveNet cannot learn variable-centric representations. However, these models include cross-variable modeling through variable embeddings, raising questions about the necessity for an additional variable-centric approach. (Line 56-57).
   - The paper states that weighted averaging of sequence-centric and variable-centric modeling is ineffective, but no concrete explanation is provided to support this claim. (Lines 61-62)

   **b) Lack of Motivation for MoE Adoption**: The paper does not adequately explain why MoE is suitable for traffic prediction (Lines 63-72).

   **c) Scope Mismatch**: A significant portion of the paper is dedicated to the limitations of spatio-temporal prediction approaches, yet the paper only focuses on traffic prediction.

2. **Insufficient Challenge Identification:** Suboptimal routing at the early stage of training, as mentioned in line 78, is a well-known issue with MoE models, and numerous solutions have been proposed over the years. As a paper aiming to adapt MoE to traffic prediction, there lacks in-depth thinking about domain-specific challenges, making the paper's contribution limited.

3. **Limited Method Novelty and Unclear Interpretation:**

   **a) Sequence-centric design:** The sequence-centric experts used in this work are similar to previous approaches [1, 2]. However, the paper neither provides a clear motivation for using these experts nor highlights how they differ from homogeneous ST-MoE approaches [3]. In addition, the relevant refs [2, 3] are not cited or compared.

   **b) Variable-centric design:** The variable-centric expert design closely resembles the ideas proposed in itransformer [4], which is not cited or compared. Moreover, itransformer was initially designed for time series forecasting, and it may not be well-suited for modeling dynamic spatial dependencies as discussed in [5].

   **c) Gating network design:** The motivation for altering the classical sparse gating network into the proposed form in Eqn. (8) is not sufficiently explained. Furthermore, the paper suggests that relationships between nodes are often sparse in urban traffic flow forecasting (Lines 295-297), which contradicts Eqns. (3) and (4) that model global dependencies among all nodes.

4. **Insufficient experiment:**

   **a) Unconvincing baselines results:** The paper simply copies the baselines results from ref [1]. Differences in computing resources and the absence of repeated experiments with varying random seeds cast doubt on the reliability of the results.

   **b) Lack of in-depth comparison with homogeneous MoE models [3].**

   **c) Concerns with the model efficiency:** MoE models are typically known for their efficiency, but TITAN falls short in this regard. As shown in Table 3, the inference time of TITAN is even longer than that of GWNet, raising concerns about its practical applicability.

## Reference

[1] Hyunwook Lee,  et al. TESTAM- A Time-Enhanced Spatio-Temporal Attention Model with Mixture of Experts. ICLR2024.

[2] Wenzhao Jiang, et al. Interpretable Cascading Mixture-of-Experts for Urban Traffic Congestion Prediction. KDD 2024

[3] Shuhao Li, et al. ST-MoE: Spatio-Temporal Mixture-of-Experts for Debiasing in Traffic Prediction. CIKM 2023.

[4] Yong Liu, et al. iTransformer: Inverted Transformers Are Effective for Time Series Forecasting. ICLR 2024.

[5] Zezhi Shao, Exploring Progress in Multivariate Time Series Forecasting: Comprehensive Benchmarking and Heterogeneity Analysis. TKDE 2024.

**Questions:**

1. In lines 246-247, you mention that 'all sequence-centric models share the same inductive bias, which limits the performance of the MoE model.' The three sequence-centric experts are heterogeneous with differing inductive biases. Furthermore, why does shared inductive bias constrain the performance?

2. In lines 250-252, you state that 'the variable-centric model does not share the same hidden state structure.' Could you provide evidence or a theoretical explanation for why this characteristic complicates control by the MoE routing mechanism?
3. How does the annealing routing method perform compared with other advanced MoE routing strategies?
4. How would TITAN likely perform in different spatio-temporal applications beyond those already tested in your study?

---

### Official Review · Reviewer_gMeq · 2024-11-03

**Soundness:** 3
**Presentation:** 2
**Contribution:** 2
**Rating:** 3
**Confidence:** 5

**Summary:**

TITAN offers an innovative approach to traffic prediction by addressing the limitations of sequence-centric models, which often miss variable-centric interactions. TITAN bridges this gap through a combination of sequence-centric, variable-centric experts, and a prior knowledge-centric expert.

The model’s prior knowledge-centric strategy supervises routing, enhancing accuracy, while an expert annealing strategy reduces leader reliance during training for better adaptability. Empirical results show TITAN outperforms the state-of-the-art.

**Strengths:**

1. The paper introduces two innovative components: a variable-centric modeling for traffic forecasting and a prior knowledge-centric modeling in the gating mechanism to anneal the overfitting/suboptimal problem.

2. The ablation study effectively demonstrates the impact of these components on traffic speed prediction tasks.

3. The paper includes comprehensive comparative experiments on two traffic speed datasets with details. It helps reproduce the work.

**Weaknesses:**

1. The novelty presented in Section 3 is limited, as much of the content describes existing methods. More emphasis is needed on detailing the unique contributions of this work. What is the difference in the variable-centric modeling between your work and [2]?

2. The paper is not well organized. For example, in the introduction section, you mentioned that current studies focus on sequence-centric modeling rather than variable-centric modeling. Since this serves as the motivation for your research, it is essential to clearly define what sequence-centric modeling and variable-centric modeling entail. Additionally, you should explain why a variable-centric approach is important. One figure can help explain it.

3. The Figure 1 is unclear. The prior section needs to be redone to better illustrate its connection with the memory component. Additionally, the representation of hidden states should be clearer, as the current color scheme makes it difficult to distinguish whether the hidden states in the routing process come from a variable-centric or sequence-centric approach. Furthermore, clarification is needed regarding whether the two sets of QKV (query, key, value) weights are shared or independent. Lastly, the output appears to be isolated from the MoE routing section, which raises concerns about the cohesiveness and interaction between these components.

4. Traffic flow typically refers to the number of vehicles passing along a specific road segment. To develop a model for traffic flow data, it is recommended to conduct experiments using established datasets such as PEMS03, PEMS04, and LargeST. To demonstrate that your model is effective for general traffic forecasting tasks, it is advisable to validate its performance across a variety of datasets that include not only traffic flow but also traffic density/occupancy data.

5. Some results in Table 1 are from other papers, like TESTAM, MegaCRN [1]. It is necessary to claim it in the paper.

6. More case studies can help show the contribution of your work. For example, you can provide examples of challenging cases where the inclusion of the prior knowledge-guided gating mechanism leads to significant model performance improvements.

[1] Lee H, Ko S. TESTAM: A Time-Enhanced Spatio-Temporal Attention Model with Mixture of Experts[C], ICLR 2024.

[2] Haotian Gao, Renhe Jiang, Zheng Dong, Jinliang Deng, Yuxin Ma, and Xuan Song. Spatial-TemporalDecoupled Masked Pre-training for Spatiotemporal Forecasting, April 2024.

**Questions:**

Same as the weakness.

---

### Note · Authors · 2024-12-22

I have read and agree with the venue's withdrawal policy on behalf of myself and my co-authors.